# Rapid Therapeutic Drug Monitoring of Voriconazole Based on High-Performance Liquid Chromatography: A Single-Center Pilot Study in Outpatients

**DOI:** 10.3390/antibiotics14050474

**Published:** 2025-05-08

**Authors:** Satoru Morikawa, Yusuke Yagi, Moemi Okazaki, Narika Yanagisawa, Tomoaki Ishida, Kohei Jobu, Takumi Maruyama, Takahiro Kato, Miyuki Matsushita, Yu Arakawa, Yuka Yamagishi, Yukihiro Hamada

**Affiliations:** 1Department of Pharmacy, Kochi Medical School Hospital, 185-1 Kohasu, Oko-town, Nankoku 783-8505, Japan; satoru.morikawa.rp@hitachi-hightech.com (S.M.); jm-moeoka@kochi-u.ac.jp (M.O.); jm-narika@kochi-u.ac.jp (N.Y.); jm-kouheij@kochi-u.ac.jp (K.J.); jm-maruyama.takumi@kochi-u.ac.jp (T.M.); jm-takkato@kochi-u.ac.jp (T.K.); hamada_yukihiro@kochi-u.ac.jp (Y.H.); 2Department of Chromatography Sales, Hitachi High-Tech Analysis Corporation, Tokyo 105-6409, Japan; miyuki.matsushita.zy@hitachi-hightech.com; 3Department of Infection Prevention and Control, Kochi Medical School Hospital, Nankoku 783-8505, Japan; y-arakawa@kochi-u.ac.jp (Y.A.); y.yamagishi@mac.com (Y.Y.); 4Department of Clinical Pharmacy, Graduate School of Pharmaceutical Sciences, Nagoya City University, Nagoya 467-0001, Japan; t-ishida@phar.nagoya-cu.ac.jp; 5Department of Clinical Infectious Diseases, Kochi Medical School, Kochi University, Kochi 780-8072, Japan

**Keywords:** voriconazole, rapid therapeutic drug monitoring, high-performance liquid chromatography, adverse events, outpatient care

## Abstract

**Background/Objectives:** Voriconazole (VRCZ) use requires accurate monitoring to avoid suboptimal drug levels and adverse effects. In addition, the appearance of resistant fungal strains is a problem that needs attention. Blood concentration measurement is the monitoring technique of choice; however, it is slow, limiting its clinical application. This study aimed to evaluate the clinical utility of rapid therapeutic drug monitoring (TDM) for VRCZ using high-performance liquid chromatography with ultraviolet detection (HPLC-UV) compared to conventional outsourced liquid chromatography–tandem mass spectrometry (LC-MS/MS) testing in outpatient care. **Methods**: VRCZ blood concentrations were measured using HPLC-UV and LC-MS/MS. Reporting times, accuracy, and clinical outcomes were assessed for outpatients receiving VRCZ treatment. Safety was monitored for renal, hepatic, and visual toxicities. **Results**: HPLC-UV significantly reduced reporting times (0.433 h vs. 74.3 h, *p* < 0.001), and Deming’s regression analyses showed a strong correlation with LC-MS/MS results (Pearson’s r = 0.988). Bland–Altman analysis showed an average difference of 0.025 μg/mL between HPLC-UV and LC-MS/MS. Prospective monitoring of three outpatients revealed no adverse events, enabling safe and effective VRCZ dosing. **Conclusions**: Rapid VRCZ TDM using HPLC-UV is a cost-effective and feasible approach for outpatient care, significantly improving reporting times and patient safety. Further studies and cross-facility collaboration are needed to expand its application.

## 1. Introduction

Voriconazole (VRCZ) is widely recognized for its efficacy in treating invasive fungal infections, especially in immunocompromised patients, such as those with hematological malignancies or those undergoing organ transplants [1,2,3]. However, the challenge of inter-patient variability in VRCZ pharmacokinetics can lead to suboptimal drug levels and adverse effects, including hepatotoxicity and neurotoxicity [4,5,6]. In addition, the emergence of azole-resistant fungal strains due to long-term exposure can be a problem [7,8]. To mitigate these risks, TDM and antifungal stewardship are recommended [9,10,11,12,13,14]; however, conventional blood concentration measurement methods are often slow due to the reliance on external laboratories, limiting their timely clinical application [15]. Few studies have comprehensively assessed the clinical impact of rapid methods in outpatient settings, particularly in Japan, where the infrastructure for real-time TDM remains underdeveloped [16]. Furthermore, to enhance safety, there is a review of the drugs eligible for insurance reimbursement under specific drug treatment management programs. However, VRCZ is currently not eligible for outpatient reimbursement [17,18]. This study aims to fill this gap by directly comparing the reporting times and clinical outcomes of rapid TDM based on high-performance liquid chromatography, with the conventional external laboratory method, specifically within an outpatient care framework.

## 2. Results

### 2.1. Analysis of VRCZ Blood Concentration

Figure 1 shows the measurement flow based on method A (HPLC-UV). The methodology for LM1010 use has been optimized and incorporated into a routine procedure. After turning on the system, only one click is needed to perform all steps of the analysis, from automatic conditioning to quality control and sample measurement, allowing optimal and precise measurements. In addition, working efficiency has been improved by setting up a specific set of consumables and general reagents required for this method (Table 1). The time required for measuring the blood concentration of VRCZ by pharmacists on the day of outpatient treatment was 0.433 (0.400–0.467) h for method A and 74.3 (71.2–94.2) h for method B (LC-MS/MS) (*p* < 0.001). Method A included the time required for TDM by pharmacists on the day of outpatient treatment.

### 2.2. Validation of VRCZ Blood Concentration Analysis

Patient backgrounds are described in Table 2. The correlation between the blood concentrations of VRCZ measured using method B (x-axis) and method A (y-axis) for 24 points of 7 patients is shown in Figure 2. The results of Deming’s regression showed a slope of 1.03 (95% confidence interval [CI]: 0.95–1.10) and an intercept of −0.08 (95% CI: −0.28–0.11), and Pearson’s r = 0.988. No statistically significant proportional or constant error was observed between the two measurement methods, confirming very high correlation and accordance between the measured values. The mean difference in the quantitative bias between the blood concentrations of VRCZ using method A and B was 0.025 μg/mL, with a range of −0.5059–0.5559 μg/mL (Figure 3). The Bland–Altman analysis showed that the difference between the two measurement methods was statistically acceptable, confirming good agreement between the measured values.

### 2.3. Evaluation of Safety of VRCZ in Outpatients

Patient backgrounds are described in Table 3. In this study, two males and one female patient were monitored in a positive manner. These three patients were over 65 years old. Two of them had chronic necrotizing pulmonary aspergillosis, and one had fungal sinusitis of the ethmoid sinus. There were no problems with the combination drugs of VRCZ in any of the cases, and no renal dysfunction, hepatic dysfunction, or visual impairment due to VRCZ was observed during the treatment. Two patients continued treatment after remission and are currently being followed up; one patient discontinued treatment due to death.

## 3. Discussion

VRCZ has nonlinear pharmacokinetics and is a good candidate for TDM due to inter-racial differences in genetic polymorphisms and numerous drug–drug interactions [9,10,19,20]. However, several variables surrounding VRCZ blood concentration measurement or the therapeutic environment hinder TDM, as reported by pharmacists [16,17,21]. This study is the first to consider the expansion and improvement of outpatient care with TDM, aiming to resolve these interfering factors.

In many facilities in Japan, the measurement of voriconazole blood concentrations is currently outsourced [17]. The main reason for this is that insurance reimbursement is only approved for in-hospital testing, and as a result, the time lag in dosing design based on TDM is a clinical hurdle [17]. For this reason, attempts have been made to optimize treatment using a combination of TDM software provided free of charge by Pfizer Japan Inc. (Tokyo, Japan, version 1.2) and pharmacometrics [22,23]; however, this has not led to a fundamental solution. Regarding quantitative VRCZ methods, previous studies indicate LC-MS/MS provides high sensitivity/specificity [24] and throughput [25], but substantial costs and required expertise limit its use in medical facilities [26,27]. Bioassay, known for low cost and simplicity, exhibits poor interassay reproducibility, an inadequate dynamic range, and lengthy analysis [26,28]. Enzyme immunoassay, despite promising study results [29], lacks widespread application. HPLC-UV relies on the light absorption characteristics of VRCZ, providing the method with its key advantages: high specificity and selectivity, while minimizing interference from other drugs [30,31,32]. Simplicity and low cost represent further advantages, as the method avoids expensive equipment and resources [15]. In this study, we demonstrated the selectivity of HPLC-UV over LC-MS/MS. These advantages allow it to be put into practice in other medical facilities. In the validation study, it was confirmed that both methods had extremely high correlation and HPLC-UV did not fall outside the reported value range. The optimal concentration range for VRCZ monitoring is between 1 and 4 μg/mL [11]. Given that the Lower Limit of Quantification (LLOQ) of HPLC-UV is 0.276 μg/mL, it is sensitive enough to identify patients with concentrations below the therapeutic range, minimizing the risk of undetected sub-therapeutic levels compared to the LLOQ of LC-MS/MS. However, the Bland–Altman analysis results show an error of ±0.56 μg/mL in measurements, which may affect clinical judgment when the concentration is outside the optimal range (less than 1 μg/mL or greater than 4 μg/mL); therefore, careful interpretation is necessary. It is possible to follow the manufacturer’s methodology when using LM1010; however, caution should be practiced for certain patient groups. Further research is necessary to elucidate its effects in such special cases [33,34,35]. Sample analysis results with the LM1010 were significantly faster (0.433 h) than outsourced testing (74.3 h). It was therefore possible to provide immediate feedback on dosage using TDM. As TDM should be performed before the steady state is reached [36,37,38], it is necessary to establish a system for its rapid implementation in order to confirm early treatment effects and manage the risk of side effects. The implementation of TDM using this method, which excels in speed, will be a useful means of solving early response issues.

The treatment period for chronic pulmonary aspergillosis is often long; long-term support is therefore important, and outpatient management is also necessary [39,40]. In this study, TDM using the HPLC-UV method was performed in three outpatients with infections caused either by *Aspergillus* spp. or *Acrophialophora* spp., allowing the patients to continue safe VRCZ treatment without the occurrence of renal dysfunction, hepatic dysfunction, or visual impairment. As a result of the prospective TDM with HPLC-UV, we found it necessary to individually reduce the dosage for two of the three patients. In the case of outsourcing testing, there is a concern that the delay in optimizing the dosage using TDM may worsen safety. Furthermore, the impact on safety of not using method verified in this study without outsourcing inspections is estimated to be even greater.

Since insurance calculation cannot be made in outpatient settings when outsourcing testing is performed, there is a concern that the need for TDM is not recognized [17]. Pharmacists will need to be able to quickly confirm individual dosage regimens and safety through TDM and propose test orders to prescribing physicians.

Another concern is that the induction of resistance in *A. fumigatus* due to long-term exposure to voriconazole may lead to a deterioration in patient prognosis [41]. It is necessary to consider whether TDM can help prevent antimicrobial resistance, especially from *A. fumigatus* [42,43]. All patients included in the study had VRCZ target concentrations set at 1–4 μg/mL and were strictly monitored from the beginning of treatment. Maintaining VRCZ concentrations below 1 μg/mL from the start of treatment through to the maintenance phase triggers fungal resistance [44,45]. Therefore, the results of our rapid TDM study may provide a valuable insight for antifungal stewardship strategy and goal achievement.

The primary limitation of this study is that the sample size was relatively small, which may have limited statistical power. Therefore, the current results are preliminary and need to be verified by cohort studies using a larger sample size. The secondary limitation of this study is that it is a retrospective, single-center study. In the future, it will be necessary to conduct more prospective case–control studies involving multiple facilities, children, and patients with renal failure. Further verification of the external validity of the results is also encouraged to improve treatment outcomes, reduce medical costs, and counter the development of antimicrobial resistance. The third limitation is that it is not possible to comprehensively evaluate the environmental impact, practicality, and economic feasibility of the analytical methods, although such analysis has been attracting attention in recent years. It is necessary to evaluate the sustainability of the analytical method [46]. In Japan, as the separation of dispensing and prescribing is progressing, it is necessary to provide information to the medical facilities, hospitals and clinics, through collaboration with insurance dispensing pharmacies [18]. To further develop VRCZ TDM for outpatient care, immediate sharing of test results using this HPLC-UV and cross-facility TDM collaboration is necessary. The future challenge will be to build evidence and share it within the community.

## 4. Materials and Methods

### 4.1. Analysis of VRCZ Blood Concentration

The method for measuring the blood concentration of VRCZ in the hospital was based on high-performance liquid chromatography, using LM1010 high-performance liquid chromatograph (Hitachi High-Tech Analysis Corporation, Tokyo, Japan). Hereafter, the LM1010 high-performance liquid chromatograph is referred to as LM1010. LM1010 was used to measure the blood concentration of VRCZ according to the operating procedure manual provided with the manufacturer-recommended VRCZ method. This was defined as method A. First, a standard filter method allows the purification of residual samples of >400 μL using a 1 mL syringe (TERUMO, Tokyo, Japan) connected to a 0.45 μm syringe filter (Hitachi High-Tech Analysis Corporation, Tokyo, Japan) followed by the obtention of more than 150 μL filtered samples. Second, solid phase extraction using Spin columns (Hitachi High-Tech Analysis Corporation, Tokyo, Japan) was employed for sample purification (the detailed flow was provided in https://doi.org/10.1186/s40780-021-00225-8), then the eluted sample was transferred to an auto-sampler installed in LM1010 (the available validation data were provided in https://doi.org/10.1186/s40780-021-00225-8). Samples were prepared by two or more people, and operator variability was addressed through systematic training and the creation of a standard operating procedure. LM1010 allows the use of HPLC-UV assay for voriconazole using a calibration curve. We only needed one measurement cycle of 7 min. We used the evaluation system for LM1010 (LaChrom LM Type A separating column integrated) in this study. The QC solution, a standard solution used to confirm system compatibility, was supplied in ampoules provided by Hitachi High-Tech Analysis. In addition, standard solutions for calibration curves were prepared using ampoules provided by Hitachi High-Tech Analysis Corporation, and the calibration curves were created using the single-point absolute calibration method. No internal standard substances were used.

The outsourced measurement of VRCZ blood concentration was carried out using the LC-MS/MS method with a high-performance liquid chromatograph tandem quadrupole mass spectrometer from SRL Inc. This was defined as method B. After the removal of proteins, the sample was separated using HPLC (ODS column) and detected using a liquid chromatograph mass spectrometer (Shimadzu Corporation, Kyoto, Japan). The instruments and LC-MS/MS analytical conditions are provided in https://www.shimadzu.co.kr/sites/shimadzu.co.kr/files/pim/pim_document_file/technical/white_papers/11476/jpo116066.pdf, accessed on 9 April 2025.

### 4.2. Validation of Analysis of VRCZ Blood Concentration

The study participants were inpatients who had received treatment with VRCZ injections and oral medication in Kochi Medical School Hospital. The blood concentration of VRCZ in the participants was measured using method A and B. When multiple blood samples were collected from the same patient, all available data were included in the analysis to account for intra-individual variability. Blood concentrations were compared using Deming’s regression analysis and Bland–Altman analysis [47,48]. The regression coefficient in Deming’s regression analysis was expressed as the slope and intercept, and Pearson’s r. The limits of agreement for the error in Bland–Altman analysis were expressed as the mean value ± 1.96 × standard deviation of the difference in the measurement results for method A and B. Repeated measurements from the same patient were treated as individual data points while recognizing the potential correlation within subjects.

### 4.3. Evaluation of the Safety of VRCZ in Outpatients

Outpatients given oral VRCZ at Kochi Medical School Hospital between January and December 2024 were included in the study. The exclusion criteria included patients who had opted out. The following patient information was collected: patient age, sex, underlying disease, biochemical and hematological parameters, microbiological tests performed, fungal species detected, type of concomitant drug, VRCZ dosage, duration of treatment, and VRCZ blood concentrations using method A and B. The workflow for measuring blood concentrations of VRCZ in outpatients within the hospital is shown in Figure 4. Outpatients who were the subjects of this study did not take their medication on the day of their visit because blood sampling for concentration measurement is performed before VRCZ administration. VRCZ was administered approximately one hour after arrival at the hospital. The difference between method A and method B is whether the measurement is performed in-house or outsourced. In method A, pharmacist-conducted TDM is performed before the patient leaves the hospital, whereas in method B, pharmacist-conducted TDM is performed at the patient’s next visit. The TDM target range for VRCZ blood concentrations was set at 1–4 μg/mL for all patients [11]. The patients enrolled in this study were followed up prospectively during the treatment period, and the presence or absence of renal dysfunction, hepatic dysfunction, and visual impairment were monitored. Based on the diagnostic criteria of acute renal failure of the Acute Kidney Injury Network [49], renal dysfunction was diagnosed if the serum creatinine level increased 0.3 mg/dL above the baseline. Based on the International Consensus Meeting diagnostic criteria [50], liver dysfunction was diagnosed if the alanine aminotransferase level was double the normal upper limit, and the onset of hepatic dysfunction was diagnosed if the alkaline phosphatase level exceeded the normal range. Visual impairment was defined as either temporary photophobia or subjective symptoms of color vision changes, or a diagnosis of decreased visual acuity or photopsia after consulting an ophthalmologist [51].

### 4.4. Statistical Analysis

Categorical data are reported as percentages and compared using Fisher’s exact test (two-tailed). Continuous data are reported as median [IQR] and compared using the Mann–Whitney U test. All statistical analyses were performed with EZR version 1.29 (Saitama Medical Center, Jichi Medical University, Saitama, Japan). Statistical significance was set at *p* < 0.050. Deming’s regression analysis and Bland–Altman analysis were performed using the R statistical software version 4.4.3 (https://www.r-project.org/).

### 4.5. Ethical Approval

This study was approved by the Ethical Review Board of Kochi University School of Medicine (registration number: ERB-109876, approval date: 20 February 2024).

## 5. Conclusions

Conclusively, this study shows that HPLC-UV methodology for TDM is significantly faster than conventional LC-MS/MS, making TDM for outpatients more efficient, and improving VRCZ use in outpatients. It is suggested that the rapid determination of blood concentration of VRCZ by HPLC-UV will be an effective tool for TDM implementation and medical care support. We believe that our findings will lead to the widespread use of rapid and appropriate VRCZ TDM in outpatient care and will become a framework for many medical facilities.

## Figures and Tables

**Figure 1 antibiotics-14-00474-f001:**
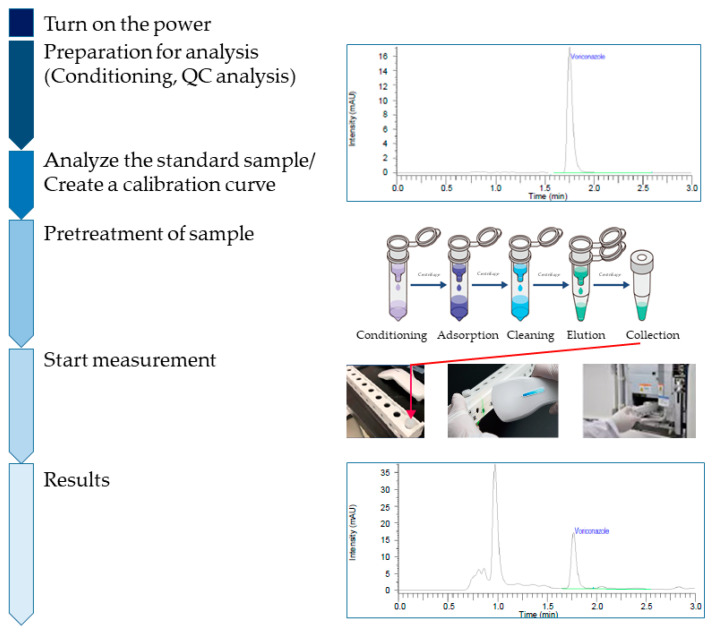
The HPLC-UV (method A) methodology workflow. The details of the measurement of VRCZ blood concentration using the HPLC-UV system. HPLC-UV, high-performance liquid chromatography with ultraviolet detection; VRCZ, voriconazole.

**Figure 2 antibiotics-14-00474-f002:**
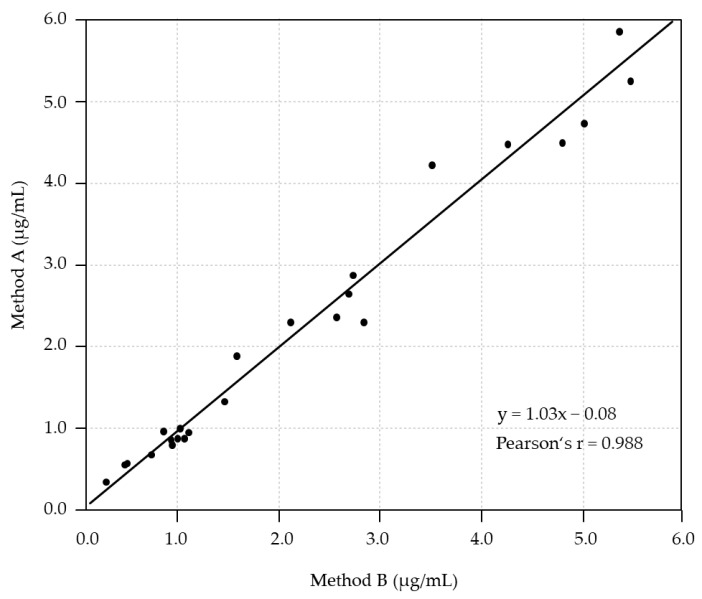
Deming’s regression: Comparison of VRCZ blood concentrations between method A vs. B. Deming’s regression of VRCZ blood concentrations between method A (*n* = 24) and B (*n* = 24). The regression coefficient in Deming’s regression analysis was expressed as the slope and intercept. VRCZ, voriconazole.

**Figure 3 antibiotics-14-00474-f003:**
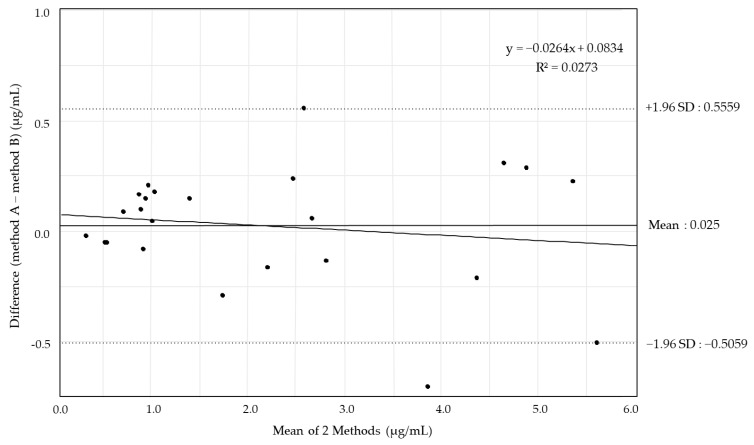
Bland–Altman analysis: The agreement between method A and B for VRCZ measurement. The limits of agreement for the error in Bland–Altman analysis were expressed as the mean value ± 1.96 × standard deviation of the difference in the measurement results for method A (*n* = 24) and B (*n* = 24). VRCZ, voriconazole.

**Figure 4 antibiotics-14-00474-f004:**
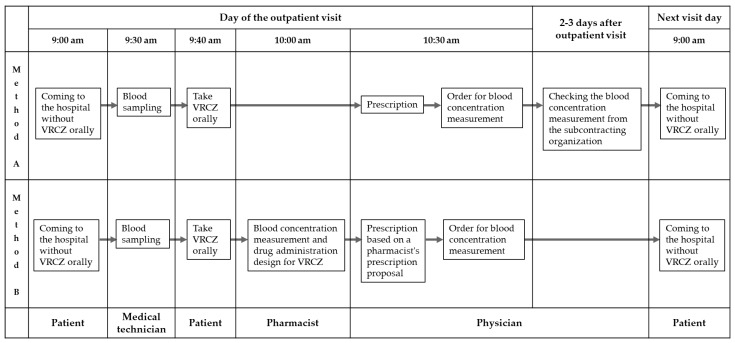
Process for measuring outpatient VRCZ blood concentrations in-hospital. Flow of TDM based on VRCZ blood concentration results in outpatients (method A vs. method B). VRCZ, voriconazole.

**Table 1 antibiotics-14-00474-t001:** Chromatographic conditions for VRCZ determination.

	LM1010 (Method A)
Instrument	LM1010 (certified medical device), Hitachi High-Tech Analytical Science (Tokyo, Japan)
Mobile phase	Mobile phase A, mobile phase B, Hitachi High-Tech Analytical Science
Column	LaChrome LM Type A, Hitachi High-Tech Analytical Science
Extraction method	Spin column set, Hitachi High-Tech Analytical Science
Plasma volume (μL)	150
Fresh plasma sample	1 μg/mL spiked serum recovery: 99.6%
Frozen (−30 °C) plasma sample	1 μg/mL spiked serum recovery: 97.3%
Lower limits of quantitation (μg/mL)	0.276
Calibration curve range (μg/mL)	1–5
CV (%)	0.630–1.02
Accuracy (%)	99.6–104.2
Retention time of VRCZ blood concentration (min)	1.76

VRCZ, voriconazole; CV, coefficient of variation.

**Table 2 antibiotics-14-00474-t002:** Baseline patient data for VRCZ analysis validation.

		*p*-Value
Number	7	N/A
Age	64 (55–76)	N/A
Sex (male) (%)	2/7 (28.6)	N/A
Weight (kg)	52.0 (36.5–67.0)	N/A
ALT (U/L)	17.0 (5.0–71.0)	N/A
AST (U/L)	25.5 (13.0–69.0)	N/A
Scr (mg/dL)	0.78 (0.36–1.02)	N/A
BUN (mg/dL)	18.0 (6.0–40.9)	N/A
Total bilirubin (mg/dL)	0.30 (0.20–1.4)	N/A
VRCZ initial dosage (mg/day)	600 (300–600)	N/A
VRCZ maintenance dosage (mg/day)	300 (150–400)	N/A
VRCZ blood concentration (μg/mL)		
Method A	1.78 (0.33–5.61)	0.750
Method B	1.53 (0.31–5.47)	
Time required for measuring (h)		
Method A	0.433 (0.400–0.467)	<0.001
Method B	74.3 (71.2–94.2)	

VRCZ, voriconazole; ALT, alanine transaminase; AST, aspartate transaminase; Scr, serum creatinine; BUN, blood urea nitrogen.

**Table 3 antibiotics-14-00474-t003:** Outpatient VRCZ treatment: Patient characteristics.

Patients No.	No.1	No.2	No.3
Age (years)	71	70	65
Sex	Female	Female	Male
Weight (kg)	36.5	63.9	50.5
Diagnosis	Chronic necrotizing pulmonary aspergillosis	Fungal sinusitis	Chronic necrotizing pulmonary aspergillosis
Pathogen	*Aspergillus niger*	*Acrophialophora* spp.	*Aspergillus fumigatus*
Concomitant medication (dosage/day)	Levothyroxine Sodium Hydrate 62.5 μg	Butyric acid-producing bacillus 3 g	L-Carbocisteine 1500 mg
	Ambroxol Hydrochloride 45 mg	Lansoprazole 30 mg	Dimemorfan Phosphate 30 mg
	Montelukast Sodium 10 mg	Amlodipine Besilate 5 mg	Entecavir Hydrate 0.5 mg
	Gefapixant Citrate 90 mg	Lemborexant 2.5 mg	Eszopiclone 1 mg
		Ramelteon 8 mg	
VRCZ dosage (/day)	400 mg → 280 mg → 200 mg	600 mg → 300 mg	600 mg → 400 mg → 300 mg
VRCZ blood concentration (μg/mL)	4.52 → 2.40 → 1.99 → 2.74	2.29 → 2.51	5.26 → 4.37 → 2.17
VRCZ treatment period	9 months~	4.5 months	5 months~
Hepatic function (before treatment)	ALT: 9 U/L, AST: 30 U/L	ALT: 33 U/L, AST: 17 U/L	ALT: 16 U/L, AST: 19 U/L
Hepatic function (after treatment)	ALT: 10 U/L, AST: 22 U/L	ALT: 17 U/L, AST: 22 U/L	ALT: 15 U/L, AST: 26 U/L
Renal function (before treatment)	Scr: 0.39 mg/dL, BUN: 8.5 mg/dL	Scr: 0.43 mg/dL, BUN: 18.4 mg/dL	Scr: 0.68 mg/dL, BUN: 7.3 mg/dL
Renal function (after treatment)	Scr: 0.53 mg/dL, BUN: 15.6 mg/dL	Scr: 0.54 mg/dL, BUN: 18.0 mg/dL	Scr: 0.57 mg/dL, BUN: 7.1 mg/dL
Hepatic dysfunction	N/A	N/A	N/A
Renal dysfunction	N/A	N/A	N/A
Visual impairment	N/A	N/A	N/A
Treatment outcome	Remission	Death	Remission

VRCZ, voriconazole; ALT, alanine transaminase; AST, aspartate transaminase; Scr, serum creatinine; BUN, blood urea nitrogen.

## Data Availability

The original contributions presented in this study are included in the article. Further inquiries can be directed to the corresponding author.

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
