# Peer review of "Rapid Therapeutic Drug Monitoring of Voriconazole Based on High-Performance Liquid Chromatography: A Single-Center Pilot Study in Outpatients"

_antibiotics, 2025, doi:10.3390/antibiotics14050474_

Round 1
Reviewer 1 Report
Comments and Suggestions for Authors
The manuscript by Morikawa et al. showcases a study on feasibility of application of an in-house HPLC-UV method instead of an outsourced LC-MS/MS approach for more timely therapeutic drug monitoring of the antimicrobial agent voriconazole. The underlying theme of this work is valuable and commendable in that the analysis results should be reported as quickly as possible when dealing with diseases which show time-dependent progression, as a delay in dose optimisation may hinder treatment outcomes. The manuscript provides valuable insight into therapeutic drug monitoring practices for voriconazole and the reported patient results may be of interest to other scientists working in this area. I am confident this manuscript will be suitable for publication in Antibiotics, after adding some more details to the methodology and discussion sections. My comments are as follows:
- Method details about the HPLC-UV protocol should be added to the manuscript if possible. Namely, the composition of the mobile phases, the dimensions and chemistry of the HPLC column, the isocratic or gradient elution composition, flow rate and temperature, analysis duration. This information is not visible in Figure 1, Table 1 nor in the Materials & methods section. The authors refer to a manufacturer-recommended VRCZ protocol - this should be cited, e. g. using a link to a site that shows all the necessary data, and a short description should be added to the manuscript.
- Details on sample preparation are also not fully described – such as the chemistry of the sorbent, its dimensions and mass, dilution applied to the samples before extraction (if any), solvents used for the extraction, use of sample preconcentration, type of solvent used for the final sample which was subjected to analysis... Either a reference to a published protocol or a brief description should be added if possible.
- If possible, method validation should also be described in slightly more detail: were calibration curves and QC samples prepared as standard solutions or spiked plasma samples? How were all the necessary solutions prepared – used solvents, concentrations of stock solutions, calibrant levels, manner of spiking of plasma samples if it was performed? Was any kind of internal standard used? Again, if these results are already published elsewhere, a reference with a brief description is sufficient.
- Some basic information on the used LC-MS/MS method should be stated in the manuscript if possible – column chemistry and dimensions, mobile phase composition, analysis duration, use of internal standards, sample preparation method details…
- The authors state that the main merit of the new method is in its speed of output, i. e. the duration of time between the analysis and the reporting of the results, which is indeed of great importance and one of the key aspects of this work. However, the HPLC-UV method may also have other advantages in regard to LC-MS/MS, both in an ecological and an economical sense, which can further add to the strength of the discussion section. “Greenness” and “blueness” analyses using published open-source software such as AGREE and BAGI, or at least adding a short paragraph on these method properties, might be insightful and increase the depth of the comparison.
- In Deming regression, I suggest the authors add 95 % confidence intervals of the slope and the intercept as those are also useful indicators of the methods’ similarity.
- The interpretation of the cross-validation results should also be elaborated in more detail in the text, in regard to the limits of acceptability for each tested parameter.
- Figure 4 is not entirely clear – I suggest including the times of drug administration. A brief description of this figure in the manuscript text may also be beneficial, especially as it appears that this figure is not cited in the text. Additionally, the endpoints and rationale for TDM should be briefly described to make Table 3 easier to follow: what are the target concentrations, at what time after dosing are the concentrations measured…
Titles of the tables and figures can be improved as there are duplicated sentences. The number of samples may be added to the descriptions of figures 2 and 3. The abbreviations should be properly introduced - e.g. the detection method should also be written when introducing the HPLC-UV and LC-MS/MS abbreviations.
Reviewer 2 Report
Comments and Suggestions for Authors
The study presents a promising approach to rapid TDM but requires methodological rigor and larger cohorts to support its claims. Methods are appropriate but underdescribed. Results are internally valid for the cohort but lack external robustness. Data interpretation overstates safety conclusions due to limited follow-up. Therefore, revision should focus on transparency, statistical justification, and clinical context.
- The validation cohort (n=7) and safety evaluation (n=3) are extremely limited. Such small sample sizes reduce statistical power and generalizability. The authors should acknowledge this limitation and consider expanding the cohort or conducting a power analysis to justify feasibility.
- Critical details of the HPLC-UV method are missing (e.g., mobile phase composition, gradient program, column temperature). Include these to ensure reproducibility. Additionally, clarify whether samples were analyzed in duplicate and how inter-operator variability was addressed.
- The LLOQ for HPLC-UV (0.276 µg/mL) is stated, but LC-MS/MS’s LLOQ is omitted. Compare sensitivity between methods and discuss clinical implications (e.g., risk of undetected sub-therapeutic levels with HPLC-UV).
- The spin column extraction method lacks validation data (e.g., recovery rates, extraction efficiency). Include this to ensure analytical reliability, especially for frozen samples stored beyond one week (stated as “applicable” without evidence).
- The introduction highlights azole resistance but the study does not explore how rapid TDM might mitigate this. Discuss theoretical mechanisms (e.g., avoiding subtherapeutic levels) and align findings with stewardship goals.
Round 2
Reviewer 2 Report
Comments and Suggestions for Authors
The author have addressed my concerns, therefore, I recommend to accept it in current form.